# TurboRAG: Accelerating Retrieval-Augmented Generation with Precomputed KV Caches for Chunked Text

## Abstract

Current Retrieval-Augmented Generation (*RAG*) systems concatenate and process numerous retrieved document chunks for prefill which requires a large volume of computation, therefore leading to significant latency in time-to-first-token (*TTFT*). To reduce the computation overhead as well as TTFT, we introduce *TurboRAG*, a novel RAG system that redesigns the inference paradigm of the current RAG system by first pre-computing and storing the key-value (*KV*) caches of documents offline, and then directly retrieving the saved KV cache for prefill. Hence, online computation of KV caches is eliminated during inference. In addition, we provide a number of insights into the mask matrix and positional embedding mechanisms, plus fine-tune a pretrained language model to maintain model accuracy of TurboRAG. Our approach is applicable to most existing large language models and their applications without any requirement in modification of models and inference systems. Experimental results across a suite of RAG benchmarks demonstrate that TurboRAG reduces TTFT by up to 9.4x compared to the conventional RAG systems (on an average of 8.6x), but reserving comparable performance to the standard RAG systems.

## 1 Introduction

Retrieval-augmented generation (RAG) systems have been emerged as a promising direction to alleviate some challenges faced by large models (LMs), e.g., hallucinations (Mallen et al., 2023; Khandelwal et al., 2020; Izacard et al., 2022). As shown in Figure 1a that large-scale documents in these systems are typically segmented into a myriad of short document chunks that can be embedded for retrieval. Upon the arrival of a user-input query, the most relevant chunks are then retrieved and prepended to the input as an augmented query fed to an LM for prefill, followed by decoding in an autoregressive (AR) manner to generate responses. RAG system effectively utilizes factual documents as supplementary data to enhance model's ability to generate more accurate and contextually rich responses, hence widely adopted by various applications, such as question answering (Siriwardhana et al., 2023; Han et al., 2024) and content creation (Khattab et al., 2022), etc. However, existing RAG systems come with several limitations from the system perspective.

First, repeatedly recalled document chunks require recomputation of the key-value (KV) caches, leading to redundant computation. Second, the augmented document contains substantially more tokens for prefill which contributes to considerably more computational overhead since the computation cost of KV caches is quadratic to the input sequence length. It, hence, significantly increases TTFT, making RAG systems possibly unsuitable for applications that have stringent constraints on response time. Third, as a side effect of the requirement in substantial computation resources for concatenated document prefill, the batch size on a single device might be limited.

The fundamental reason for these issues lies in prefill paradigm of the current RAG system, which involves online computation of the concatenated long documents, i.e. it collects the most relevant documents and then performs prefill for them together. A natural question arises: *can we alter this paradigm to remarkably reduce the computation overhead of prefill?* If we were able to precompute the KV caches of the retrieved documents offline and let the prefill stage directly uses these saved KV caches to rebuild the complete KV cache for a request online, a large body of online computation can

then be completely eliminated, thus significantly reducing system's TTFT and improving inference efficiency. This essentially transforms the RAG's prefill stage into a hybrid paradigm combining both offline and online processing. Compared to the conventional RAG system, the only issue is that the transformation may result in inconsistent attention mask matrix and position IDs. Resolving these inconsistencies would yield an efficient RAG solution.

In this paper, we propose TurboRAG, which is grounded in two observations. First, as illustrated in Figure 2a, cross-attention among different documents is exceedingly sparse in RAG models and the text contents between most documents are actually independent. Second, for relative position embedding techniques, such as RoPE(Su et al., 2024), only the relative distance between two positions matters. Consequently, the relative positional embeddings of a document are equivalent no matter the KV cache is computed using the individual document or the entire concatenated documents. Inspired from these observations, TurboRAG first pre-computes and stores the KV caches for each document offline. It then injects the relevant KV caches of the retrieved documents into a user request to construct the complete KV caches for prefill using the independent attention mask matrix from the Figure 2c and the standard RoPE.

Compared to the conventional RAG system, experimental results across the LongBench multi-document QA benchmarks demonstrate that TurboRAG reduces TTFT by up to 9.4x and on an average of 8.6x, with comparable accuracy to the baseline. Simultaneously, during online inference, TurboRAG reduces computational resource utilization by $98.46\%$ compared to standard RAG, which significantly increases the maximum supported batch size and enhances throughput. Additionally, regression experiments indicate that TurboRAG does not exhibit any significant degradation in other general capabilities compared to standard RAG.

In summary, we make three major **contributions**. First, we design a novel pipeline that decomposes the prefill stage of conventional RAG systems into offline and online phases to notably reduce the overhead of KV cache computation. Second, we propose simple yet effective techniques to handle attention mask and position IDs so that model accuracy is maintained. Third, we achieve a substantial improvement of 9.4x in TTFT over the state-of-the-art multi-document QA benchmarks without compromising accuracy.

## 2 RELATED WORK

Retrieval-Augmented Generation (RAG) (Lewis et al., 2020) has achieved significant progress in natural language processing by integrating large language models (LLMs) with external knowledge databases. This integration enhances the ability of generative models to produce accurate, relevant, and context-rich responses. Recent studies (Borgeaud et al., 2022; Jiang et al., 2024; Trivedi et al., 2022; Ram et al., 2023) have demonstrated that RAG significantly outperforms pure generative models across various benchmarks, thereby gathering considerable amounts of research interests in various domains such as question answering (Siriwardhana et al., 2023; Han et al., 2024), code generation (Lu et al., 2022), and content creation (Khattab et al., 2022), etc. However, as a relative new research topic, the current RAG systems still suffer from some drawbacks, among which low performance and long latency are the most prominent ones. Addressing these problems would effectively make RAG more applicable to latency-sensitive LLM tasks.

As illustrated in Figure 1a, the workflow of a naive RAG system comprises two steps: retrieval and generation, combining offline preparation with online processing to enhance performance. In the offline phase, RAG utilizes embedding models such as BGE (Chen et al., 2024a)) and GTE (Li et al., 2023) to convert external knowledge sources (e.g., document chunks) into high-dimensional vectors, which are then indexed into a specialized vector database. Upon receiving a user request, RAG first accesses this vector database to perform a similarity search, retrieving documents that best match the request based on semantic content. Subsequently, RAG integrates the content of these retrieved documents with the original user request to form an augmented query, which is input into the LLM to generate a more informative and contextually relevant response (Topsakal & Akinci, 2023).

Researchers have proposed various methods to optimize the performance of retrieval-augmented generation (RAG) systems. Some approaches modify the attention computation mechanism to reduce computational complexity (Wang et al., 2020; Choromanski et al., 2020; Monteiro et al., 2024;

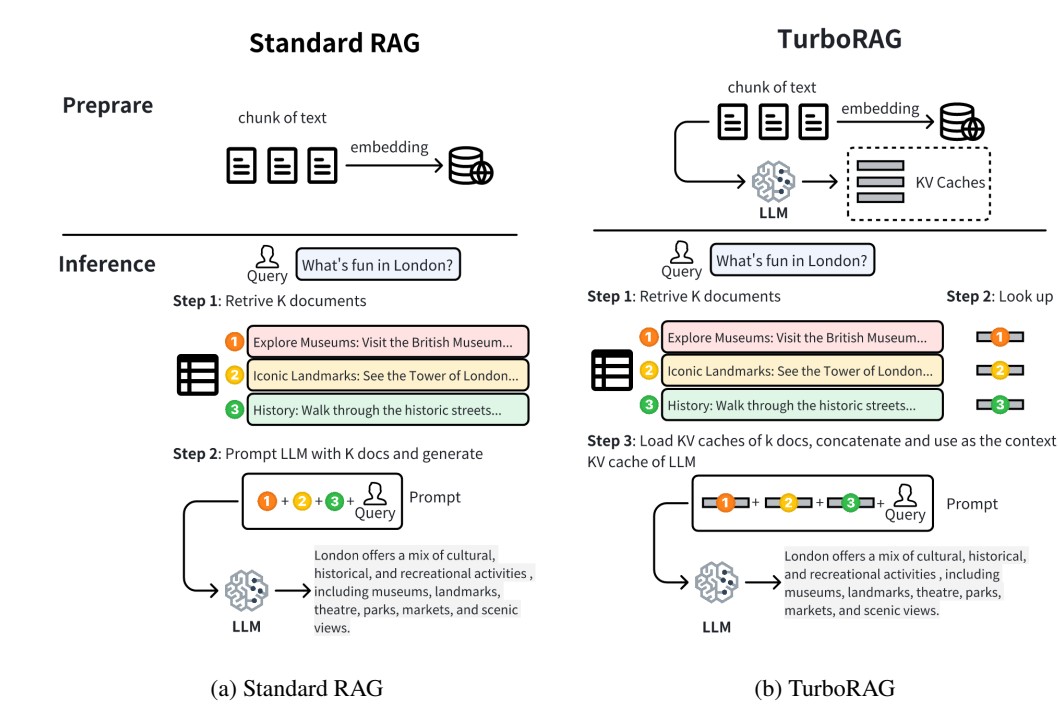

Figure 1: Pipeline of Standard RAG and TurboRAG. TurboRAG pre-compute the KV cache for each chunk of text and reuse during RAG inference.

Choromanski et al., 2020; Kitaev et al., 2020), serving as general optimizations for the model architecture. Furthermore, FiD (Fusion-in-Decoder) (Hofstätter et al., 2023) independently processes each retrieved passage through the encoder, limiting self-attention to individual passages. This ensures that the computational cost scales linearly with the number of passages. The decoder then aggregates the retrieved information, allowing the model to better extract relevant support from multiple retrieved passages. Parallel Context Windows (PCW) (Ratner et al., 2022) addresses long-text processing by dividing texts into smaller chunks and restricting attention computations within chunks. While this method avoids expensive cross-window attention, it does not resolve position embedding discontinuities, making it better suited for tasks like RAG where windows are relatively independent. Sparse context selection (Zhu et al., 2024) further accelerates RAG inference by adding a LLM-based filtering mechanism to reduce the number of retrieved documents processed, significantly enhancing efficiency in large-scale retrieved documents.

Additional techniques focus on compressing and merging KV caches, as well as distributed inference, to reduce computational overhead in processing long sequences (Wang et al., 2024; Liu et al., 2024; Zhang et al., 2024). While effective for general long-text generation, these methods face challenges in RAG systems due to the dynamic nature of retrieved passages, where directly concatenating cached states can lead to accuracy drops. Multi-level caching systems like RAGCache (Jin et al., 2024) optimize efficiency by reusing intermediate states across queries. However, RAGCache stores KV caches for identical queries that frequently appear in historical dialogue records, relying on exact matches between contexts and prompt text. This approach faces two main challenges: (1) it cannot handle variations in the order of recalled documents; (2) it suffers from a hit rate issue, requiring recalculation when discrepancies occur between the cached context and the current prompt.

To address the performance issues, we propose *TurboRAG*, a novel RAG optimization scheme by precomputing and storing the key-value (KV) caches of document fragments offline. During online generation, the model directly utilizes these precomputed KV caches, avoiding redundant computation of the retrieved document fragments. To be best of our knowledge, this is the first work in the literature that attempts to redesign inference paradigm of the current RAG system by transforming

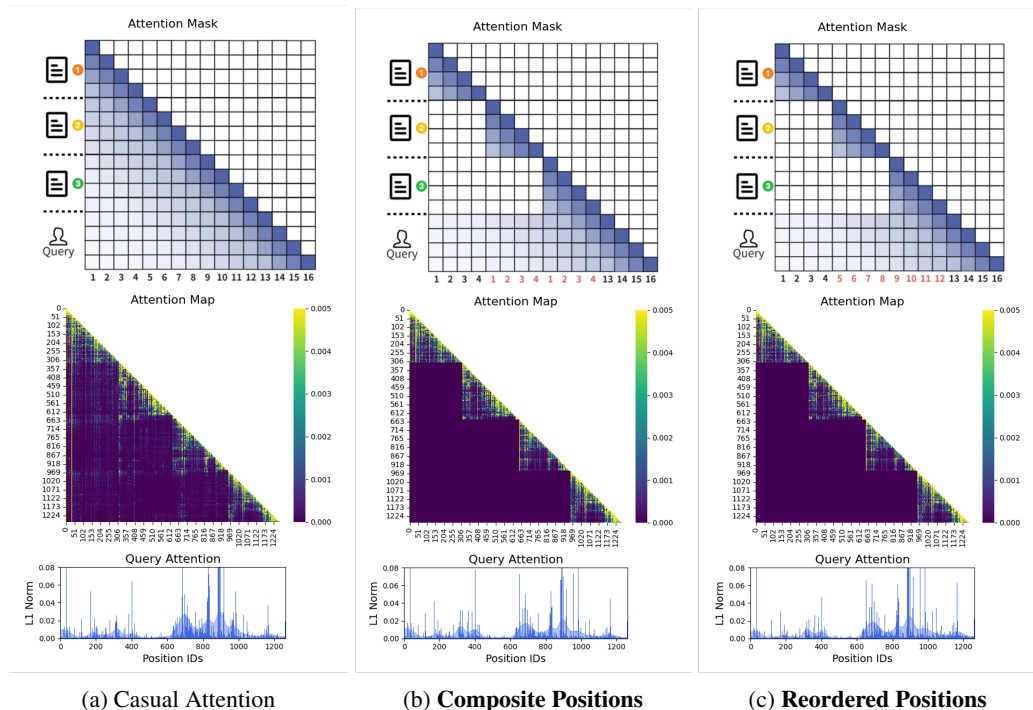

(a) Casual Attention        (b) **Composite Positions**        (c) **Reordered Positions**

Figure 2: The first row presents three distinct setting of attention mask matrices and position IDs. (a) Lower triangular casual attention, where the entire context is attended to. (b) **Independent Attention** and **Composite Positions**, which use the original position IDs for each chunk. (c) **Independent Attention** and **Reordered Positions**, where each document can only attend to itself and rearrange the position IDs for tokens in chunk to standard monotone increasing numbers. In the second and third rows, we present an instance of RAG to visualize and analyze the distribution of the attention matrices under different settings, as well as the distribution of attention scores from the query to the context chunks. This instance consists of four text chunks and a user query, as detailed in Appendix A. In the standard setting shown in the first column of second row, it can be observed that the attention scores between different chunks are quite sparse; each document primarily focuses on its internal information. Furthermore, in the third row, the distribution of attention scores from the query to the context chunks indicates that even when the attention between documents is fully masked, the distribution of attention scores from the query to the documents does not exhibit significant variation, remaining concentrated in the documents that contain relevant information.

the online computation of KV caches for the retrieved documents into offline processing. This approach significantly reduces the computational complexity of the RAG systems and could become a powerful enabler for LLM applications that have restricted latency constraints.

## 3 METHODOLOGY

This section presents TurboRAG, a novel approach to improve the performance of conventional RAG systems without sacrificing accuracy. We formalize the problem in Section 3.1 and discuss the differences in the attention mask matrix and position IDs between TurboRAG and existing RAG systems in Section 3.2. Section 3.3 explains how we trained the model to adapt to the new attention mask matrix and position IDs. We introduce the TurboRAG inference pipeline in Section 3.4.

### 3.1 PROBLEM FORMALIZATION

Conventionally, given a user query $q$, we retrieve top $k$ document chunks, $[c_1, \ldots, c_k]$, and send them to a LLM that sequentially generates the textual outputs. We denote the number of tokens in $x$ as $\text{len}(x)$ and we assume $len(c_i) = l$. In existing RAG, we first compute the prefill using

$q$ and the concatenated $c$, denoted as a concatenated context sequence $[c_1, \ldots, c_k, q]$, to obtain the corresponding hidden states $\boldsymbol{X}^c$. At each decoding step $t$, the model computes attention scores based on $\boldsymbol{X}^c$. Let $\boldsymbol{X} = [\boldsymbol{X}_1, \boldsymbol{X}_2, \ldots, \boldsymbol{X}_t]$ be the hidden states of the tokens generated so far, where $\boldsymbol{X}_t$ is the hidden state for the current token being generated. The model computes the query $\boldsymbol{Q}_t$, key $\boldsymbol{K}_i$, and value $\boldsymbol{V}_i$ matrices for context at position $i$:

$$\boldsymbol{Q}_t = \boldsymbol{X}_t \boldsymbol{W}_Q, \quad \boldsymbol{K}_i = \boldsymbol{X}_i^c \boldsymbol{W}_K, \quad \boldsymbol{V}_i = \boldsymbol{X}_i^c \boldsymbol{W}_V \tag{1}$$

Here, $\boldsymbol{W}_Q$, $\boldsymbol{W}_K$, and $\boldsymbol{W}_V$ are the learned weight matrices. The attention score is computed using the dot product of the query and the key, scaled by the square root of the dimension of the key vectors $d$:

$$\text{Attention\_scores} = \frac{\boldsymbol{Q}_t \boldsymbol{K}_i^T}{\sqrt{d}} \tag{2}$$

For RoPE, it is necessary to multiply $\boldsymbol{Q}_t$ and $\boldsymbol{K}_i$ by their corresponding position embedding separately as shown in Equation 3:

$$\boldsymbol{Q}_t^{'} = \begin{pmatrix} q_0 \\ q_1 \\ q_2 \\ q_3 \\ \vdots \\ q_{d-2} \\ q_{d-1} \end{pmatrix} \oplus \begin{pmatrix} \cos t\theta_0 \\ \cos t\theta_0 \\ \cos t\theta_1 \\ \cos t\theta_1 \\ \vdots \\ \cos t\theta_{d/2-1} \\ \cos t\theta_{d/2-1} \end{pmatrix} + \begin{pmatrix} -q_1 \\ q_0 \\ -q_3 \\ q_2 \\ \vdots \\ -q_{d-1} \\ q_{d-2} \end{pmatrix} \oplus \begin{pmatrix} \sin t\theta_0 \\ \sin t\theta_0 \\ \sin t\theta_1 \\ \sin t\theta_1 \\ \vdots \\ \sin t\theta_{d/2-1} \\ \sin t\theta_{d/2-1} \end{pmatrix} \tag{3}$$

where $\theta_m = 10000^{-2m/d}$. A benefit of this equation is that the position embedding for $\boldsymbol{Q}$ and $\boldsymbol{K}$ can be computed independently. Furthermore, the final result of the multiplication of the two position embeddings is solely dependent on the positional difference between them. Since this is an autoregressive model, we need to apply a causal mask to ensure that the model does not attend to future tokens. This is typically achieved by multiplying with a lower triangular masking matrix:

$$\text{Attention\_scores} = \text{Attention\_scores} * \boldsymbol{M} \tag{4}$$

where $\boldsymbol{M}$ is the masking matrix. $\boldsymbol{K}^{'}$ and $\boldsymbol{V}$ are generally referred to as *KV cache*, which is stored for the subsequent computation of attention scores in the later regressive decoding. The attention scores are then normalized using the *softmax* function to obtain attention weights. Finally, the output for the current token is computed as a weighted sum of the value vectors.

## 3.2 Position ID Rearrangement

This section presents the technique we developed to ensure that the concatenated KV cache computed offline for each document is as effective as the KV cache computed using the whole originally retrieved documents. Figure 2 illustrates the differences in the attention mask matrix and position IDs between the two methods.

The online concatenation of the KV cache requires that there is no cross-attention between multiple document chunks during inference, which is a significant distinction from the lower triangular mask matrix employed by the current RAG system. We denote this new attention modality in Figure 2c as **Independent Attention**, which effectively simulates the scenario of retrieving the KV caches and concatenating them. As illustrated in Figure 2c, cross-attention between documents are all set to zero, and when decoding the answer, attention scores are computed among query, answer and all documents.

Another issue arising from TurboRAG is the computation of position embeddings. The key cache computed for each $c_i$ are denoted as $\boldsymbol{K}^{c_i}$. If the KV caches are simply concatenated, all $\boldsymbol{K}^{c_i}$ will consist of position IDs ranging from 0 to $l$. Consequently, the finally combined IDs will be represented as $[0, \ldots, l, 0, \ldots, l, 0, \ldots, l]$, which we refer to as **composite positions**. This presents a problem: when decoding at step $t$, the positional difference between an element in $\boldsymbol{K}^{c_i}$ and $t$ does not correspond to the actual token index difference. For instance, the third element in $\boldsymbol{X}^{c_2}$ at this point has a positional difference of $t-3$, while the actual token index difference should be $t-(l+3)$.

To resolve this issue, we rearrange the positions of all key cache to obtain $[0, \ldots, l, l+1, \ldots, 2l, 2l+1, \ldots, k \cdot l]$. We refer to this new positions arrangement as **reordered positions**. Equation 3 demonstrates that RoPE can effectively support **reordered positions**; it suffices to retain the $\boldsymbol{K}$ and $\boldsymbol{V}$ from Equation 1 when saving the KV cache. After concatenating KV caches, we can compute the key cache $\boldsymbol{K}^{'}$ using Equation 3 with the new position IDs, which is quite straightforward. For $\boldsymbol{Q}$, we can leverage Equation 3 to get $\boldsymbol{Q}^{'}$ using its position ID, which is the same as the standard RAG system.

However, the new attention mask matrix and position embedding could lead to a significant accuracy drop in question-answering tasks. To mitigate this issue, we need to specifically train the model to make the LLM be able to handle this new setting. To compare the effects of different positional indices, we will conduct experiments on both **reordered positions** and **composite positions** in Section 4. Next, we will introduce the training details.

### 3.3 Adapting LLMs for Precomputed Cache Concatenation

In order to enable a pretrained LM to execute diverse instructions, it is a common practice to fine-tune the LM using a pile of specifically created instruction learning data that encompasses various instruction tasks. For example, we usually need specialized data to enhance the reading comprehension capability used in a RAG model. Instruction learning data is generally constructed in the following format to train the model.

> You are an accurate and reliable AI assistant capable of answering questions by referencing external documents. Please note that the external documents may not always be related to the question. The documents are as follows:
> `<|doc_start|>{chunk_1}<|doc_end|>`
> `<|doc_start|>{chunk_2}<|doc_end|>`
> `<|doc_start|>{chunk_3}<|doc_end|>`
> ...
> If the information in the documents contain the correct answer, you will provide an accurate response. If the documents do not contain the answer, you will refuse to answer.
>
> Question: {`que`}

Standard supervised fine-tuning (SFT) typically employs the attention mask matrix and position embeddings shown in Figure 2a to fine-tune the LM using the data with the above format. However, to make sure that the pretrained LM can accommodate to new patterns exhibited in the mask matrix and position embedding during inference, TurboRAG used the mask matrix and position embedding in Figure 2b and Figure 2c to fine-tune the LM. After the fine-tuning, the LM would be able to see the same context KV cache produced from training while conducting inference. Therefore, it would not experience the accuracy regression in question-answering tasks.

### 3.4 The TurboRAG Pipeline

With the fine-tuned LLM, the inference pipeline of TurboRAG is enumerated as follows (Figure 1b):

1. **Document Encoding (offline)**: The documents are encoded into embedding vectors using a transformer-based model like Bert(Devlin et al., 2019). These document embeddings are stored in a vector index to facilitate efficient similarity search.

2. **Document Prefill (offline)**: Use an LLM to perform prefill offline. It computes the KV caches for each document and saves them in the database.

3. **Query Encoding**: The input query is encoded into a vector using the same Bert model.

4. **Retrieval**: The encoded query is used to perform a similarity search in the vector database to retrieve the most relevant documents.

5. **Contextual KV cache Formation (online)**: Retrieve the stored KV cache corresponding to the documents and concatenate them in the way demonstrated in Figure 2. The combined KV cache forms a comprehensive context for the query.

6. **KV Cache Prefill (online)**: The LLM processes prefill using the combined KV caches for the input query.

7. **Response Generation (online)**: After the prefill phase is accomplished, the LLM starts to generate the response and return to the user.

It is evident that the usage process of TurboRAG is fundamentally consistent with that of standard RAG, making it highly convenient to use. We will be releasing the modified implementation code as open source.

## 4 EXPERIMENTS

This section evaluates performance and accuracy of a number of TurboRAG model variants against the conventional RAG models. Specifically, we seek to answer the questions below in this section:

- How does TurboRAG perform on document question-answering (QA)?
- What is the overall TTFT performance of TurboRAG compared against the Näive RAG system on popular benchmarks?
- How large is the regression in the general capabilities of TurboRAG models?
- How efficient is TurboRAG in scaling inference batch sizes?

### 4.1 EXPERIMENT SETUP

We selected gpt-4o-2024-08-06 as the baseline due to its excellence in many benchmark suites. For brevity, we refer the conventional RAG system as "Näive RAG". We also fine-tuned two models for TurboRAG, namely TurboRAG-composite and TurboRAG-reordered corresponding to **composite positions** and **reordered positions**, respectively. All three models are fine-tuned on a dataset composed of 50% document QA data and 50% general tasks (e.g., code, dialogue, reasoning). All data are publicly accessible. For a detailed composition of the dataset, please refer to Appendix B.

**Training Setup** We base our training on Qwen2-7B(Yang et al., 2024), performing SFT on the aforementioned dataset. The fine-tuning was conducted on 32 NVIDIA A100 80GB GPUs with a batch size of 256 sequences, using a learning rate of 1e-5 and the AdamW optimizer(Loshchilov, 2017). Both Näive RAG and TurboRAG models were trained using the same data proportions to ensure comparability.

### 4.2 DOCUMENT QA ACCURACY

Let's first evaluate the accuracy of document QA via intensive study on RGB Benchmark(Chen et al., 2024b), a bilingual benchmark designed to test a model's ability to answer questions on retrieved documents. We followed the testing methodology provided by the official guidelines and let each query extract five documents during the evaluation. In addition, we also measured the accuracy with varying noise levels from 0.2 to 0.8 (e.g., *Noise Ratio* = 0.6 means 3 out of 5 retrieved documents are irrelevant or noisy). In order reveal the effectiveness of fine-tuning, we gauged accuracy of each TurboRAG configuration with and without fine-tuning.

As shown in Table 1, without fine-tuning, the accuracy drops significantly. Particularly, as the task difficulty increases (i.e., with a higher noise ratio), the accuracy can decline by nearly 20%. This is because the RAG models never learned the behavior of the new independent attention and composite positions employed in inference. Nonetheless, simply fine-tuning the model with the small dataset enables the TurboRAG models to attain impressive accuracy. Compared to the Näive RAG, even without fine-tuning, **independent attention** and **reordered positions** only decrease the average accuracy by 5.8% (96.8 vs 91.0) and 4.2% (96.8 vs 92.6). After fine-tuning, TurboRAG-reordered and TurboRAG-composite can effectively maintain the benchmark accuracy gap within 1% compared to the Näive RAG. They also demonstrated comparable performance to GPT-4o across both Chinese and English datasets even under high-noise conditions. This highlights the effectiveness of the proposed modifications in preserving high accuracy when leveraging KV cache in document QA tasks. Additional experimental data on RGB can be found in Appendix C, which also includes details on the multi-document integration tasks in the RGB dataset. The results show that even for

Table 1: Performance comparison of different models under various noise ratios in English and Chinese in RGB.

| Chinese | | | | | |
| --- | --- | --- | --- | --- | --- |
| Model | Noise Ratio | | | | |
| | 0.2 | 0.4 | 0.6 | 0.8 | Avg. |
| GPT-4o-2024-08-06 | 98.3 | 98.0 | 96.6 | 87.7 | 95.2 |
| Naïve RAG | 99.0 | 98.0 | 96.7 | 87.3 | 95.3 |
| TurboRAG-composite w/o fine-tuning | 98.3 | 96.3 | 93.7 | 79.0 | 91.8 |
| TurboRAG-reordered w/o fine-tuning | 98.0 | 96.7 | 93.3 | 81.3 | 92.3 |
| TurboRAG-composite | 99.0 | 97.3 | 96.0 | 86.7 | 94.8 |
| TurboRAG-reordered | 98.7 | 97.3 | 96.0 | 90.7 | **95.7** |
| English | | | | | |
| Model | Noise Ratio | | | | |
| | 0.2 | 0.4 | 0.6 | 0.8 | Avg. |
| GPT-4o-2024-08-06 | 99.0 | 99.3 | 98.3 | 96.3 | 98.2 |
| Naïve RAG | 99.7 | 99.3 | 99.3 | 94.3 | 98.2 |
| TurboRAG-composite w/o fine-tuning | 98.0 | 96.3 | 91.3 | 75.0 | 90.2 |
| TurboRAG-reordered w/o fine-tuning | 98.0 | 97.3 | 90.7 | 85.7 | 92.9 |
| TurboRAG-composite | 99.3 | 98.0 | 96.7 | 92.7 | 96.7 |
| TurboRAG-reordered | 99.0 | 98.3 | 96.0 | 93.7 | **96.8** |

queries requiring information synthesis across multiple documents, TurboRAG-reordered achieves accuracy comparable to that of Näive RAG.

To validate that our method proposed techniques are also directly applicable to long text input cases, we inspected TurboRAG's accuracy on an additional long-text RAG benchmark dataset, Long-Bench(Bai et al., 2023). As shown in Table 2, TurboRAG also exhibits comparable answer accuracy to that of Naïve RAG in such use scenarios.

In all experiments, the performance of TurboRAG-composite was consistently inferior to that of TurboRAG-reordered, particularly in more challenging contexts such as LongBench. This observation further validates the necessity of maintaining the accuracy of relative positional differences in positional encoding.

Table 2: Performance of Naive RAG and TurboRAG on LongBench multi-document QA (subcategories).

| Subcategory (Metric) | Context Token | Query Token | Score | | | TTFT (ms) | | |
| --- | --- | --- | --- | --- | --- | --- | --- | --- |
| | | | Naïve | Turbo Composite | Turbo Reordered | Naïve | Turbo Reordered | Speedup |
| MuSiQue (F1) | 16349 | 18.8 | 22.12 | 23.64 | 27.37 | 1610 | 171 | **9.4x** |
| 2WikimQA (F1) | 7553 | 17.0 | 35.02 | 34.28 | 39.51 | 709 | 101 | **7.0x** |
| DuReader (Rouge-L) | 10642 | 6.0 | 34.57 | 33.37 | 33.03 | 1007 | 116 | **8.7x** |
| HotpotQA (F1) | 13453 | 20.1 | 40.21 | 35.78 | 45.28 | 1333 | 147 | **9.1x** |
| Avg. | 11999 | 15.5 | 32.99 | 31.76 | **36.29** | 1165 | **134** | **8.6x** |

## 4.3 GENERAL CAPABILITY REGRESSION

To ensure that the non-standard attention masks and position IDs usded in fine-tuning does not negatively affect the models' general capabilities, we accomplished regression tests using the Open-

Compass[1] benchmark on various mainstream tasks. As summarized in Table 3, the modifications had minimal impact on the base capabilities of the models. TurboRAG-reordered showed strong generalization across tasks, with no significant performance degradation compared to Naïve RAG.

Table 3: Regression experiments of Naïve RAG and TurboRAG. Evaluated by OpenCompass.

| Model | MMLU | TriviaQA | GSM-8K | MATH |
|---|---|---|---|---|
| Naïve RAG | 69.57 | 56.90 | 79.12 | 39.54 |
| TurboRAG-reordered | 70.73 | 56.47 | 79.45 | 40.58 |
| Sub | +1.16 | -0.43 | +0.33 | +1.04 |

## 4.4 TTFT PERFORMANCE

Now we assess the impact of TurboRAG on inference speed. All models are evaluated on the LongBench dataset, with specific focus on its multi-document QA tasks. The experiments were conducted on the Huggingface *transformers*[2] using FlashAttention2(Dao, 2023) and an NVIDIA A100 80GB GPU. As shown in Table 2, TurboRAG-reordered improves the performance of TTFT by 8.6x on average, with a peak speedup of 9.4x, compared to Naïve RAG for long-documents processing. This reduction substantiates that TurboRAG can significantly reduce TTFT, thereby enhancing user experience, and consequently enables the expansion of RAG applications to cases with stringent latency requirement. The main reason of reduction in the TTFT is that the online computation overhead of KV caches for long text is largely alleviated as TurboRAG shifts the KV cache computation for each document to offline processing.

## 4.5 BATCH SCALING

Compared to Naïve RAG, TurboRAG requires to transfer KV cache from CPU to GPU, which may introduce extra communication overhead that degrades performance measured by TTFT. To evaluate the magnitude of the communication cost, we carried out experiments under a fixed total recall text length of 8192 and a query length of 128. We gathered a series of TTFT numbers with batch size ranging from 1 to 8 in two settings. One transferred the KV cache from CPU to GPU using PCIE Gen4, while the other assumed that the KV cache was prefetched to the GPU memory thereby excluding the impact of communication. Additionally, we measured the computational load for both Naïve RAG and TurboRAG under different settings. The method for calculating computational load is detailed in Appendix D.

Table 4: Generation throughput and latency on an A100 GPU.

| Batch size | Metric | Naïve | Turbo | Speedup | Turbo w/o h2d | Speedup w/o h2d |
|---|---|---|---|---|---|---|
| 1 | TTFT (ms) | 711 | 175 | **4.1x** | 44 | **16.1x** |
| | TFLOPs | 136.36 | 2.09 | | 2.09 | |
| 2 | TTFT (ms) | 1408 | 325 | **4.3x** | 56 | **25.1x** |
| | TFLOPs | 272.72 | 4.19 | | 4.19 | |
| 4 | TTFT (ms) | 2842 | 666 | **4.3x** | 97 | **29.3x** |
| | TFLOPs | 545.46 | 8.39 | | 8.39 | |
| 6 | TTFT (ms) | 4373 | 928 | **4.7x** | 134 | **32.6x** |
| | TFLOPs | 818.20 | 12.58 | | 12.58 | |
| 8 | TTFT (ms) | 5812 | 1429 | **4.1x** | 177 | **32.8x** |
| | TFLOPs | 1090.93 | 16.78 | | 16.78 | |

---

[1]https://github.com/open-compass/opencompass
[2]https://huggingface.co/

From Table 4, it is evident that as the batch size increases, the speedup ratio (decrease in TTFT) also increases without any degradation in performance. When the batch size is small, the pressure on computational resources is insufficient, resulting in a TTFT speedup value of only 16.1x between Naïve RAG and TurboRAG. As the batch size increases, GPU becomes over-utilized for naive RAG, thus leading to substantially higher latency in TTFT compared to TurboRAG. Table 4 also illustrates that, even in scenarios requiring the transfer of the KV cache from host to device (h2d), TurboRAG still achieves a fourfold speed improvement compared to Naïve RAG. In addition, we collected the TFLOPs consumed by both the näive RAG and TurboRAG for each batch size, as shown in the Metric column of Table 9. It can be seen that TurboRAG achieves astonishingly less TFLOPs, i.e. approximately $98.46\%$ reduction compared to Naïve RAG. For shorter context lengths, we also conducted comparative TTFT tests, and the results are recorded in Appendix E. Additionally, if each text chunk contains 200 tokens, recalling and concatenating 5 segments results in a total of 1000 tokens. According to the experimental results, even with a batch size of 1, a commendable speedup of up to two times can be achieved.

## 5 LIMITATION

This section discusses some limitations this paper has that we intentionally leave as the future work to further improve.

Limitation 1: Storage overhead. TurboRAG essentially trades space for time. For example, Qwen2-7B has 28 layers, 8 KV heads and its head dimension is 128. Assuming each chunk contains 512 tokens, the KV cache size in FP16 is $2 \times 2 \times 28 \times 8 \times 128 \times 512 = 28M$. The KV cache for 1 million text chunks requires 28 TB storage. While this storage may be acceptable for small to medium-sized applications, it could pose a problem for larger applications that involve billions of document chunks. In addition, a KV cache retrieval system will be needed to provide quick access to required KV cache chunks. However, we have noticed an increasing number of works to handle KV cache compression (Wang et al., 2024; Liu et al., 2024; Zhang et al., 2024), which can effectively reduce the storage requirements and are orthogonal to our work. Integrating these KV cache compression techniques into TurboRAG will be our next direction of work. Beyond disk storage, the process of loading the KV cache from disk to memory in TurboRAG also puts pressure on memory usage. During the inference phase, if the batch size is very large and the recalled KV cache is excessive while the system memory is limited (for example, when deployed on a personal laptop), it may also impact system performance.

Limitation 2: Model fine-tuning. Another Issue is that the current pipeline still requires fine-tuning of the model, which limits its applicability and prevents it from being directly used on newly emerging state-of-the-art LLMs. We are currently exploring ways to reduce or even eliminate this dependency on fine-tuning.

## 6 CONCLUSION AND DISCUSSION

This paper presented a novel approach to training and utilizing RAG that significantly reduces the time required for prefill computations when concatenating retrieved text fragments. Other techniques such as KV cache compression are orthogonal to our method, hence can be directly used to reduce latency and ease storage pressure. Our work raises a interesting question in whether cross-attention between different fragments is truly necessary. If three individuals have a piece of information, and I (Q) interact with each person (K) to obtain their information (V), and then integrate these three pieces into a complete response, would this be sufficient? The three individuals might not need to communicate with each other. Furthermore, in the inference process for long texts, many computation of cross-attention might also be redundant.

Another intriguing point is the role of positional embedding. In experiments that extend context window of LLM via position interpolation, LLMs initially are pretrained with a short context length and then continued training with a small amount of data using a longer context length. This enables the model to interpolate positions and learn two sets of position embeddings. In our work, we also exposed the model to two different sets of positional embeddings, demonstrating LLM's strong adaptability to various positional embeddings.

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

## A  DOCUMENT Q&A EXAMPLE

| Query | When is the premiere of 'Carole King & James Taylor: Just Call Out My Name'? |
|---|---|
| **Document 1** | Duke capped off a remarkable season by beating UCF 30-13 on Wednesday in the Military Bowl — the program's first bowl win since 2018. With the win, Duke got to nine wins for the first time since 2014. Mike Elko has done one of the best coaching jobs in the country in his first season with the Blue Devils. The program was barely competitive in David Cutcliffe's final seasons on the job, going a combined 5-18 (1-17 ACC) in his final two years. With Wednesday's win, Duke finished the season 9-4 overall with a 5-3 mark in ACC play. It was just the third season in school history that the Blue Devils had finished with a winning conference record and won a bowl game. Washington: After going 4-8 in 2021, Washington capped off a tremendous turnaround by beating Texas 27-20 in the Alamo Bowl. With the win, Washington finished the season with 11 wins — the most it has had in a season since 2016. That's the year the Huskies reached the College Football Playoff... |
| **Document 2** | Personal PreferencePreference is a 1987 board game created by Donal Carlston that involves guessing the order in which a player prefers foods, activities, people, and other items compared to one another. The game was published by Broderbund in the United States, Playtoy Industries in Canada, and Parker Brothers International in Britain.updated version by the original creator was launched on Kickstarter on May 1, 2023. The new version contains updated cultural references and new categories.1987 Versiongame contains cards in four categories: Food & Drink, Activities, People, and Potpourri (miscellaneous). Each card has a photo or drawing on each side and text indicating what that side represents (e.g., chocolate éclairs, climbing a mountain, Harrison Ford, spy novels). Each round, one player draws four cards from one category, or one from each category, depending on the player's position on the board. Each card is placed in a colored quadrant of the board... |
| **Document 3** | However, the concert tour took place in honor of the 40th anniversary. The two might have aged since they first performed together but neither Carole King nor James Taylor have lost a beat in all these years!The concert film includes the following songs:(You Make Me Feel Like) A Natural WomanSomething in the Way She MovesSo Far AwayCarolina in My MindCountry RoadSmackwater JackWhere You Lead (lyrics changed up as the city they're playing in replaces New York)Your Smiling FaceBeautifulShower The PeopleWay Over YonderSweet Baby James (this kicks off the second half of the film)Up on the RoofIt's Too LateFire and RainI Feel the Earth MoveYou've Got a Friend-How Sweet It Is (To Be Loved by You)You Can Close Your EyesMexico (end credits)DIRECTOR: Frank MarshallFEATURING: Carole King, James Taylor, Danny Kortchmar, Peter Asher, Russ Kunkel, Leland SklarADDITIONAL MUSICIANS: Andrea Zonn, Arnold McCuller, Kate Markowitz, Robbie KondorCarole King & James Taylor: Just Call Out My Name premiered January 2, 2022, at 9:00pm ET/PT on CNN. The film will be available on demand via cable/satellite systems, CNNgo platforms, and CNN mobile apps, beginning Monday, January 3, through Sunday, January 16. |
| **Document 4** | I was also raised to see the correlation between life and the game of football and how the process of preparation leads to success in both." Jason earned a bachelors in history, government and philosophy at Adams State in 2005, and a masters in criminal justice administration from the University of Phoenix in 2007. He added a second master's in educational methods from the University of Tulsa in 2012. He was a defensive coordinator at the University of Montana, a co-defensive coordinator at Adams State, a defensive coordinator at Valdosta State and the Colorado School of Mines, a defensive advisor at Temple University, served as a defensive assistant at Oklahoma State for two years — after a two-season stay with fellow FBS program Tulsa as outside linebackers coach... |

## B  DATA PROPORTIONS

Table 5: Sampling Ratios of Different Data Types during Model Fine-tuning

| Data Type | Sampling Ratio |
|---|---|
| Document Q&A | 50% |
| General Dialogue | 25% |
| Reasoning | 10% |
| Code | 10% |
| Others | 5% |

Table 6: Specific Data and Quantities of Document Q&A

| Data Name | Language | Quantity |
|---|---|---|
| glave-rag-v1 | English | 51,153 |
| CovidQA | English | 1,519 |
| E-Manual | English | 1,186 |
| PubMedQA | English | 22,050 |
| MS Marco | English | 2,267 |
| FinQA | English | 14,268 |
| ExpertQA | English | 1,824 |
| HotpotQA | English | 17,796 |
| TechQA | English | 1,496 |
| HAGRID | English | 3,214 |
| DelusionQA | English | 1,642 |
| BioASQ | English | 4,619 |
| CUAD | English | 2,040 |
| TAT-QA | English | 29,766 |
| BaiduSTI | Chinese | 4,032 |
| DuReader | Chinese | 10,000 |
| BaiduBaike | Chinese | 13,615 |
| Wiki | Chinese | 9,265 |

## C  SUPPLEMENTARY INFORMATION FOR RGB

Table 7: Comparison of TTFT in RGB for Naïve RAG and TurboRAG.

| Model | Context Length (tokens) | TTFT (ms) | Speedup |
|---|---|---|---|
| Naïve RAG | 743 | 87 | 2.42x |
| TurboRAG | | 36 | |

Table 8: Performance comparison of different models under various noise ratios in RGB Information Integration Task.

| Chinese | | | | |
|---|---|---|---|---|
| Model | Noise 0.2 | Noise 0.4 | Noise 0.6 | Avg. |
| Naïve RAG | 50 | 46 | 29 | 42 |
| TurboRAG-composite w/o fine-tuning | 35 | 27 | 18 | 27 |
| TurboRAG-reordered w/o fine-tuning | 30 | 21 | 20 | 24 |
| TurboRAG-composite | 53 | 41 | 32 | 42 |
| TurboRAG-reordered | 56 | 44 | 32 | 44 |
| English | | | | |
| Model | Noise 0.2 | Noise 0.4 | Noise 0.6 | Avg. |
| Naïve RAG | 57 | 48 | 36 | 47 |
| TurboRAG-composite w/o fine-tuning | 40 | 27 | 27 | 31 |
| TurboRAG-reordered w/o fine-tuning | 31 | 23 | 19 | 24 |
| TurboRAG-composite | 58 | 48 | 34 | 47 |
| TurboRAG-reordered | 57 | 51 | 34 | 47 |

## D  COMPUTATIONAL LOAD CALCULATION

Here, we present the method for calculating FLOPS, while omitting the computation of lm_head due to its relatively small proportion. Let the number of input tokens be denoted as $n_{\text{input}}$ and the context length as $n_{\text{context}}$. For a LLM utilizing the Swiglu activation function, the relevant parameters include layer_num, head_num, kv_head_num, head_size, hidden_size, and intermediate_size. For each token:

- The computational cost of the QKV transformation for each layer, denoted as $C_{\text{qkv}}$, is given by:

$$C_{\text{qkv}} = 2 \times \text{hidden\_size} \times (\text{head\_num} + 2 \times \text{kv\_head\_num}) \times \text{head\_size}$$

- The computational cost of the attention mechanism for each layer, denoted as $C_{\text{attn}}$, is expressed as:

$$C_{\text{attn}} = 2 \times \text{head\_num} \times \text{head\_size} \times n_{\text{context}}$$

- The computational cost of the projection following the attention mechanism for each layer, denoted as $C_o$, is given by:

$$C_o = 2 \times \text{hidden\_size}^2$$

- The computational cost of the multilayer perceptron (MLP) for each layer, denoted as $C_{\text{mlp}}$, can be represented as:

$$C_{\text{mlp}} = 2 \times 3 \times \text{hidden\_size} \times \text{intermediate\_size}$$

Therefore, the total computational cost can thus be expressed as:

$$\text{FLOPS} = n_{\text{input}} \times \text{layer\_num} \times (C_{\text{qkv}} + C_{\text{attn}} + C_o + C_{\text{mlp}})$$

# E    COMPARATIVE TTFT ANALYSIS FOR DIFFERENT CONTEXT LENGTHS

Table 9: TTFT (ms) for different context lengths and batch sizes on an A100 GPU.

| Seq Length | Query Length | Batch Size | Naïve | Turbo |
|------------|--------------|------------|--------|--------|
| 256 | 128 | 1 | 44.00 | 41.62 |
| 256 | 128 | 2 | 68.19 | 195.96 |
| 256 | 128 | 4 | 127.19 | 165.73 |
| 256 | 128 | 8 | 242.31 | 120.62 |
| 512 | 128 | 1 | 59.16 | 37.16 |
| 512 | 128 | 2 | 101.84 | 47.58 |
| 512 | 128 | 4 | 205.61 | 133.14 |
| 512 | 128 | 8 | 398.18 | 179.94 |
| 1024 | 128 | 1 | 97.89 | 48.79 |
| 1024 | 128 | 2 | 186.02 | 89.08 |
| 1024 | 128 | 4 | 359.95 | 139.70 |
| 1024 | 128 | 8 | 711.19 | 189.81 |

