# OpenReview forum: "TurboRAG: Accelerating Retrieval-Augmented Generation with Precomputed KV Caches for Chunked Text"
_ICLR.cc/2025/Conference — ICLR 2025 Conference Withdrawn Submission_

### Official Review · Reviewer_VvUh · 2024-11-01

**Soundness:** 2
**Presentation:** 3
**Contribution:** 2
**Rating:** 5
**Confidence:** 4

**Summary:**

The paper proposes caching key-value (KV) pairs for all documents within a Retrieval-Augmented Generation (RAG) system to reduce the overhead of KV cache computation during inference. Additionally, it introduces simple techniques for reordering position embeddings to optimize performance.

**Strengths:**

1. The paper is well-written and easy to follow, making the proposed techniques accessible to readers.

2. The proposed pipeline for RAG demonstrates a significant decrease in time to first token (TTFT), showcasing practical improvements in efficiency.

**Weaknesses:**

1. While the idea of precomputing the KV cache to reduce TTFT is effective, it is not particularly novel. Prior work, such as  [1], has already explored precomputing KV caches to achieve similar objectives. Although this paper mentions RAGCache, it lacks a thorough discussion that differentiates the proposed methods from existing approaches. Including a detailed comparison with RAGCache in both methodology and experimental results would strengthen the paper’s contribution.

2.  While precomputing document KV caches can effectively reduce TTFT, it increases storage costs, as each document needs a separate set of KV caches for different models. It is important for the paper to mention this storage issue so that readers can understand the potential trade-offs involved.

[1] RAGCache: Efficient Knowledge Caching for Retrieval-Augmented Generation

**Questions:**

N/A

---

> ### Author Response · Authors · 2024-11-28
>
> We would like to express our gratitude to the reviewer for your valuable feedback.
>
> > While the idea of precomputing the KV cache to reduce TTFT is effective, it is not particularly novel. Prior work, such as [1], has already explored precomputing KV caches to achieve similar objectives. Although this paper mentions RAGCache, it lacks a thorough discussion that differentiates the proposed methods from existing approaches. Including a detailed comparison with RAGCache in both methodology and experimental results would strengthen the paper’s contribution.
>
> RAGCache is quite different from our approach, as it stores complete key-value caches of identical queries that frequently appear in historical dialogue records (corresponding to contexts formed by the same recalled document chunks). Therefore, compared to TurboRAG, it faces two challenges. First, if the order of the recalled documents differs, RAGCache cannot be used directly. In contrast, TurboRAG has additional processing for position embeddings, allowing it to handle variations in the order of recalled documents. Second, RAGCache has a hit rate issue; it can only recall contexts that exactly match those in the index along with the prompt text, requiring recalculation for any discrepancies. TurboRAG does not face this issue, ensuring a consistent speedup. We will include this discussion in the related work section.
>
> > While precomputing document KV caches can effectively reduce TTFT, it increases storage costs, as each document needs a separate set of KV caches for different models. It is important for the paper to mention this storage issue so that readers can understand the potential trade-offs involved.
>
> For the storage issue, please refer to the overall comments C1.

---

### Official Review · Reviewer_KBLU · 2024-11-01

**Soundness:** 3
**Presentation:** 2
**Contribution:** 3
**Rating:** 6
**Confidence:** 3

**Summary:**

The authors of this paper propose the TurboRAG framework, which allows documents to be encoded offline by an LLM and used later for any retrieval augmented generation task. Their main contribution is the idea that a model can be fine-tuned in order to enhance its robustness to missing sections of the KV-cache, since the retrieved document KV-caches will be independent of each other (retrieved documents will not be able to attend to each other directly). This allows for a 5 to 10 times speedup in terms of the "time-to-first-token" without any significant performance degradation in both RAG and standard generation tasks.

**Strengths:**

- The paper is well written and clear.
- Allowing LLMs to pre-compute the KV cache of a document for later use could significantly speed-up RAG applications and reduce their computational requirements.
- Their fine-tuning methodology is simple, intuitive and apparently quite effective in allowing models to leverage independently retrieved KV-cache information.
- Their approach leads to no performance degradation in tasks outside of RAG.

**Weaknesses:**

- It is unclear whether the efficiency improvements are significant when the retrieved documents are short (efficiency improvements are only measured with 8-16k tokens. I believe that most RAG settings will be mostly working with shorter prompts than that. For example, the supporting passages in multi-hop QA datasets within LongBench are actually only around 200 tokens each.
- The metrics presented in the results sections are not well specified.
- The table column and row titles are, especially for Table 2 and 3, need to be properly capitalized and formatted.

**Questions:**

- As referred to in the weaknesses, how long must the retrieved prompt be in order to offset the latency created by moving the KV cache from the CPU to the GPU? It seems that it went from 10x to 4x with only a difference of around 4k tokens.
- Did you have any experiments on using the KV cache representations to enhance retrieval itself? It seems awkward to have to encode all documents with two models offline.
- Is there any reason why you would expect the Naive RAG model would underperform Turbo reordered RAG model in english LongBench QA tasks? This seems very strange to me.
- The fact that HotpotQA and DuReader are part of the fine-tuning data to enable TurboRAG and also the main experimental setting is makes it harder to tell if the method can truly generalize. Are we at least fully certain that there is no data leakage?

---

> ### Author Response · Authors · 2024-11-28
>
> We appreciate that the reviewer understands and recognizes the contributions of this work.  We address the main concerns as follows.
>
> > It is unclear whether the efficiency improvements are significant when the retrieved documents are short (efficiency improvements are only measured with 8-16k tokens. I believe that most RAG settings will be mostly working with shorter prompts than that. For example, the supporting passages in multi-hop QA datasets within LongBench are actually only around 200 tokens each.
>
> Please refer to the overall comments C2. Based on your feedback, we have added the experimental data, which can be found in Table 7 and Table 9 in our new version.  If each text chunk contains 200 tokens, recalling and concatenating 5 segments results in 1000 tokens. According to the experimental results in Table 9, even with a batch size of 1, we can achieve a commendable speedup of up to two times.
>
> | Seq Length | Query Length | Batch Size | Base (ms) | Cache (ms) |
> |---------------------------|--------------|------------|-----------|------------|
> | 256                       | 128          | 1          | 44        | 41.62      |
> | 256                       | 128          | 2          | 68.19     | 195.96     |
> | 256                       | 128          | 4          | 127.19    | 165.73     |
> | 256                       | 128          | 8          | 242.31    | 120.62     |
> | 512                       | 128          | 1          | 59.16     | 37.16      |
> | 512                       | 128          | 2          | 101.84    | 47.58      |
> | 512                       | 128          | 4          | 205.61    | 133.14     |
> | 512                       | 128          | 8          | 398.18    | 179.94     |
> | 1024                      | 128          | 1          | 97.89     | 48.79      |
> | 1024                      | 128          | 2          | 186.02    | 89.08      |
> | 1024                      | 128          | 4          | 359.95    | 139.7      |
> | 1024                      | 128          | 8          | 711.19    | 189.81     |
>
>
> > The metrics presented in the results sections are not well specified.
>
> Are you referring to the LongBench scores? We have added an explanation regarding this aspect.
>
> > The table column and row titles are, especially for Table 2 and 3, need to be properly capitalized and formatted.
>
> Thank you very much for your reminder; we have addressed this issue.
>
> > Did you have any experiments on using the KV cache representations to enhance retrieval itself? It seems awkward to have to encode all documents with two models offline.
>
> This is a very interesting idea, as it could reduce the number of models. A potential issue is that the cost of calculating similarity using the LLM's KV cache can be quite high. We forsee a signficant improvement in retrieval if the similarity computation overhead can be reduced.
>
> > Is there any reason why you would expect the Naive RAG model would underperform Turbo reordered RAG model in english LongBench QA tasks? This seems very strange to me.
>
> Thank you for the comment. One possible reason that the test set recalls many documents, but many of them are unrelated. The computation of cross-attention may lead to mutual interference, which therefore leads to the accuracy drop.
>
> > The fact that HotpotQA and DuReader are part of the fine-tuning data to enable TurboRAG and also the main experimental setting is makes it harder to tell if the method can truly generalize. Are we at least fully certain that there is no data leakage?
>
> Please refer to the overall comments C4.

---

### Official Review · Reviewer_7nok · 2024-11-03

**Soundness:** 3
**Presentation:** 3
**Contribution:** 3
**Rating:** 8
**Confidence:** 3

**Summary:**

1. This paper introduces TurboRAG, a novel approach to improve the performance of Retrieval-Augmented Generation (RAG) systems without sacrificing accuracy.
2. A new pipeline that decomposes the prefill stage of conventional RAG systems into offline and online phases, significantly reducing the overhead of key-value (KV) cache computation.
3. Techniques to handle attention mask and position IDs to maintain model accuracy, including:


                    a.  Independent attention between document chunks.
                    b.  Rearranged position IDs for concatenated KV caches


5. A fine-tuning approach to adapt language models to the new attention and position ID patterns.
6. Substantial improvement in time-to-first-token (TTFT) performance, achieving up to 9.4x speedup (8.6x on average) over state-of-the-art multi-document QA benchmarks without compromising accuracy.
7. The authors demonstrate that TurboRAG maintains comparable accuracy to conventional RAG systems on document QA tasks, even under high-noise conditions. They also show that the approach does not significantly impact the model's general capabilities across various tasks.
8. TurboRAG's key innovation lies in precomputing and storing KV caches for document chunks offline, then directly utilizing these caches during online inference. This approach significantly reduces computational overhead and improves inference efficiency, particularly for applications with strict latency requirements.
9. The paper provides experimental results on multiple benchmarks, including RGB and LongBench, to validate the effectiveness of TurboRAG in terms of accuracy and performance. The authors also discuss the impact on batch size scaling and overall system efficiency.

**Strengths:**

*Originality:*
1. Introduces TurboRAG, a novel approach to accelerate Retrieval-Augmented Generation (RAG) systems by precomputing key-value (KV) caches for document chunks offline.
2. Proposes innovative techniques to handle attention masks and position IDs to maintain model accuracy while using precomputed KV caches.
3. Redesigns the RAG inference paradigm by transforming the online computation of KV caches for retrieved documents into a hybrid offline-online process.

**Quality:**
1. Provides a comprehensive experimental evaluation across multiple benchmarks, including RGB and LongBench.
2. Demonstrates significant performance improvements, achieving up to 9.4x speedup in time-to-first-token (TTFT) compared to conventional RAG systems.
3. Conducts thorough regression tests to ensure the proposed modifications do not negatively impact the model's general capabilities.
4. Presents detailed ablation studies on different configurations (TurboRAG-composite and TurboRAG-reordered) and analyzes their performance under various noise ratios.

**Clarity:**
1. Well-structured paper with clear sections outlining the problem, methodology, and experimental results.
Includes informative figures (e.g., Figure 1 and Figure 2) that effectively illustrate the differences between standard RAG and TurboRAG approaches.
2. Provides clear explanations of technical concepts, such as the attention mask matrix and position ID rearrangement.
Significance:
3. Addresses a critical performance bottleneck in RAG systems, potentially enabling their application in latency-sensitive scenarios.
Achieves substantial improvements in TTFT without compromising accuracy, which could have broad implications for real-world RAG applications.
4. Proposes a method that is applicable to most existing large language models without requiring modifications to the models or inference systems.
5. Reduces computational resource utilization during online inference by 98.46% compared to standard RAG, significantly increasing the maximum supported batch size and enhancing throughput.

Overall, the paper presents a novel and significant contribution to the field of RAG systems, offering a well-executed and clearly explained approach to improve their performance while maintaining accuracy. The potential impact on real-world applications and the broader applicability of the proposed techniques add to the paper's significance in the field of natural language processing and information retrieval.

**Weaknesses:**

**Limited exploration of trade-offs:**
The paper focuses primarily on the benefits of TurboRAG but does not thoroughly explore potential drawbacks or limitations. For instance: The authors do not discuss the storage requirements for precomputed KV caches, which could be substantial for large document collections. There's no analysis of how TurboRAG might impact memory usage during inference, especially for scenarios with many retrieved documents. A more balanced discussion of these trade-offs would provide a clearer picture of TurboRAG's applicability in different settings.

 **Limited comparison to other optimization techniques:** The paper primarily compares TurboRAG to a naive RAG implementation. However, it doesn't extensively compare against other recent optimization techniques for RAG systems or long-sequence processing, such as Efficient attention mechanisms (e.g., Performer, Reformer) and Other caching strategies or optimization approaches for RAG systems . A broader comparison to other RAG optimization approaches in addition to native RAG and also to other LLM architectures would help contextualize TurboRAG's contributions within the current state of the art.

 **Limited discussion of scalability:** The paper demonstrates impressive speedups, but doesn't extensively discuss how TurboRAG scales with increasing document collections at scale or query complexity. Additional experiments or analysis on scalability would strengthen the paper's claims about TurboRAG's broader applicability.

**Questions:**

1. Could the authors provide an analysis of the storage requirements for TurboRAG compared to conventional RAG systems? This information would help readers understand the trade-offs involved.

2. How does TurboRAG impact memory usage during inference, especially for scenarios with many retrieved documents? An analysis of memory consumption would provide a more complete picture of the system's efficiency.

3. How does TurboRAG compare to other recent optimization techniques for RAG systems or long-sequence processing, such as efficient attention mechanisms (e.g., Performer, Reformer) or other RAG caching and optimization strategies?

4. how does TurboRAG scale with increasing document collections or query complexity?

5. The paper focuses on a specific LLM architecture. Have the authors tested or considered how TurboRAG might apply to other popular LLM architectures? This information would be valuable for understanding the broader applicability of the approach.

---

> ### Author Response · Authors · 2024-11-28
>
> We sincerely appreciate the reviewer's constructive comments and positive acknowledgment of our work. In response to your suggestions, we have expanded the discussion in the related work section.
>
> > Limited exploration of trade-offs: The paper focuses primarily on the benefits of TurboRAG but does not thoroughly explore potential drawbacks or limitations. For instance: The authors do not discuss the storage requirements for precomputed KV caches, which could be substantial for large document collections. There's no analysis of how TurboRAG might impact memory usage during inference, especially for scenarios with many retrieved documents. A more balanced discussion of these trade-offs would provide a clearer picture of TurboRAG's applicability in different settings.
>
> For the storage issues, please refer to the overall comments C1. Additionally, we have included a discussion of the limitations in the revised version.
>
> > Limited comparison to other optimization techniques: The paper primarily compares TurboRAG to a naive RAG implementation. However, it doesn't extensively compare against other recent optimization techniques for RAG systems or long-sequence processing, such as Efficient attention mechanisms (e.g., Performer, Reformer) and Other caching strategies or optimization approaches for RAG systems. A broader comparison to other RAG optimization approaches in addition to native RAG and also to other LLM architectures would help contextualize TurboRAG's contributions within the current state of the art.
>
> Thank you for your suggestion; we will include this discussion in the related work section.
>
> > Limited discussion of scalability: The paper demonstrates impressive speedups, but doesn't extensively discuss how TurboRAG scales with increasing document collections at scale or query complexity. Additional experiments or analysis on scalability would strengthen the paper's claims about TurboRAG's broader applicability.
>
> The longbench test set we are using typically recalls 20-30 documents, averaging around 10,000 tokens, which is quite substantial. For query complexity, your concern is valid, and we will collect datasets with complex queries in our future work to assess their impact on the model. Query complexity is indeed likely to impact the accuracy of the recall, as it tests the capabilities of the recall model.
>
> > Could the authors provide an analysis of the storage requirements for TurboRAG compared to conventional RAG systems? This information would help readers understand the trade-offs involved.
>
> For the storage issues, please refer to the overall comments C1.
>
> > How does TurboRAG impact memory usage during inference, especially for scenarios with many retrieved documents? An analysis of memory consumption would provide a more complete picture of the system's efficiency.
>
> During inference, loading the KV cache requires additional memory; however, we only load the KV cache for a few document chunks at a time, which keeps memory consumption quite low. As we mentioned earlier, the KV cache for a single document chunk occupies approximately 28MB of storage. Nonetheless, this is an good suggestion, and we will include a discussion on memory consumption in the relevant section.
>
> > How does TurboRAG compare to other recent optimization techniques for RAG systems or long-sequence processing, such as efficient attention mechanisms (e.g., Performer, Reformer) or other RAG caching and optimization strategies?
>
> The Performer and Reformer introduce new attention algorithms aimed at reducing the time or space complexity of attention calculations, serving as general optimizations for the Transformer architecture. In contrast, our goal is to reduce the online computation time of LLMs in the RAG scenario (where multiple document chunks are concatenated) by leveraging a space-for-time approach, without making structural optimizations to the LLM itself. However, we agree that it is indeed important to include this comparison in the related work section. Thank you for your valuable suggestion.
>
> > The paper focuses on a specific LLM architecture. Have the authors tested or considered how TurboRAG might apply to other popular LLM architectures? This information would be valuable for understanding the broader applicability of the approach.
>
> Please refer to the overall comments C3.

---

> > ### Comment · Reviewer_7nok · 2024-12-02
> >
> > Thanks for your reply!
> >
> > 1. please publish the updated version with new items added in Related work related to my comments above.
> > 2. Reg C3, I still feel that TurboRAG has to be compared against at least 1 other different model architecture in this paper work (not in future) as its one of the basic benchmarking approaches in LLM research. that will give an idea of how consistent this approach is working.

---

> ### Author Response · Authors · 2024-12-03
>
> Thank you for your continued response. We have uploaded a new version of the paper to this platform, which includes updates on related work, a discussion of limitations, and some additional experimental results.
>
> Regarding other model architectures, we are currently training TurboRAG on LLaMA 3.1, and we will add the experimental data to this platform as soon as possible.

---

### Official Review · Reviewer_8keK · 2024-11-04

**Soundness:** 3
**Presentation:** 2
**Contribution:** 2
**Rating:** 5
**Confidence:** 4

**Summary:**

The paper introduces a kv caching computational technique that would significantly speed up (during inference time) the conventional RAG system in terms of time to first token (which would reduce user's wait time), and significantly reduce inference-time TFLOPs. In order to implement this technique, one would need to precompute the kv caches for all documents in a database. The paper makes the assumption that cross-attention between different retrieved documents are unnecessary, setting them to zero, which makes this computation technique an attention-approximation method. The paper reorder positional ids for the precomputed kv caches of the retrieved documents from a database, and using a fine-tuned LLM, it is able to achieve overall comparable performance on the RGB Benchmark as a gpt-4o powered conventional RAG system with some performance degradation in high-noise setting.
This computational technique is applicable to the following resource-constrained setting: an organization that has large storage space to store all pre-computed kv caches, has the hardware to fine-tune large language models (>=7B), has a preference to use open-sourced LLM (that use relative positions as positional embeddings) as generator rather than commercially available ones such as GPT-4o, whose user query instances and documents in database satisfy the assumption that cross-attention between documents are unnecessary for answer user queries, and is concerned about optimizing inference time.

**Strengths:**

This paper introduces a new computational technique that would speed up the inference time of an open-sourced RAG system. The paper makes the observation and subsequently the assumption that cross-attention between documents during inference is unnecessary. To avoid doing so would reduce significantly computation during inference. Therefore, this computational technique can be viewed as an attention-approximation. This assumption seems to hold for current RAG benchmarks such as RGB and LongBench multi-doc QA. This paper is written with sufficient motivation , and is able to explain its proposed computational technique with clarity. This paper identifies two technical problems with naively concatenating pre-computed kv caches, i.e. misrepresentation of positional ids, and a pre-trained LLM would suffer from distributional shift with reordered positional ids. The paper then proceeds to solve these two problems by using either composite or reordered positional ids, and then fine-tune a pre-trained LLM. In the experiment section, the paper shows that on RGB benchmark, turboRAG is only slightly worse than gpt-4o based naive RAG, and is significantly better than Qwen-7B based naive RAG. On 4 datasets from LongBench, this paper shows that turboRAG achieves 9x speedup and significant inference-time resource saving over Qwen-7B based naive RAG.

**Weaknesses:**

1. insufficient related work discussion. TurboRAG essentially uses an attention-approximation technique that assumes no cross-attention between different documents, this falls in line with many other work that manipulate and attempt to utilize the sparsity in attention maps. But few work along this line is mentioned and compared with the computational trick in turboRAG to convince me that it is a novel approach. This paper also reorders positional ids in RoPE, there are also many similar works that manipulate positional ids to various gains, such as efficiency in long-context inferencing, no such work is mentioned and compared with the positional id reordering trick used in the paper to convince me that is is a novel approach. Therefore, I would say the novelty and originality of this work is moderately low.


2. I am concerned about the limitations TurboRAG imposes on the research direction of RAG. As I wrote in the summary section, TurboRAG requires model fine-tuning which rules out most commercial LLMs. TurboRAG also requires a significant amount of offline processing and storing all precomputed kv caches, this is generally not feasible with real-life database. This paper does not discuss storage cost and time for offline kv computation. Lastly, the assumption made in paper (cross-attention between documents are unnecessary) would limit future work that adapts this technique and prevent them to explore harder, more challenging and more meaningful RAG tasks where the user query requires a careful synthesis of information in different documents, such as scientific document summarization, meta analysis generation etc. In fact, while some current benchmarks (5 explored in this paper, 4 from long bench and RBG) do not require information synthesis between different retrieved documents, this limitation in benchmarking resources should not be the cause to make provincial and limiting assumptions that would negatively impact future research efforts in RAG. FYI no limitation section is included in this work even though there is almost one spare page. Because of these limitations, I would incline to reject this paper in its current version.


3. I have some questions on the experiment evaluation sections that I will reserve to the Questions Section.

**Questions:**

1. First of all, it is unfair to use GPT-4o for a non-English benchmark because GPT-4o is predominantly trained for the English language. Second, why is the TFLOP, TTFT not reported for the RGB English dataset?

2. Why is gpt-4o's baseline not reported for the LongBench multi-doc QA datasets? I think there are plenty of space left to include these results.

3. Can the authors address the potential data contamination from its fine-tuning dataset to the 5 datasets used in the evaluation section (4 from long bench and 1 from RGB). Musique, wikimqa and hotpotqa are primarily based on wikipedia articles, and many datasets used in Table 6 are also based on wikipedia articles. Also, HotpotQA is included in table 6, why is it also used for testing?

4. I think the paper would benefit if the authors show this turboRAG technique is applicable to other LLMs, such as the more recent state-of-the-art Llama3.1 8B and 70B.

5. Currently, there is no link to code, model checkpoints and fine-tuning data. This raises issue of reproducibility.

---

> ### Author Response · Authors · 2024-11-28
>
> Thank you for the thoughtful review of our work! Please allow us to address your concerns and answer the questions.
>
> > Insufficient related work discussion
>
> Thanks for the suggestion. We have added related work on sparse attention and reordered positional IDs in the revised version.
>
> > Limitations of  model fine-tuning
>
> Thanks for pointing out. We agree that fine-tuning limits the applicability of our work and prevents it from being directly used by newly emerging state-of-the-art LLMs. We are currently exploring ways to reduce or even eliminate the dependency on fine-tuning. However, the majority of existing techniques that attempt to extend the context window length have to modify positional embedding. In order to adapt to the increased context window length, these techniques commonly tend to continue pretraining or fine-tune the foundatoin model. Inpired from these works, we hence employed fine-tuning to guarantee that TurboRAG delievers "the best" model performance.
>
> > Storage cost
>
> Please refer to overall comment C1. We have also included this part in the discussion of limitations. Thank you for your reminder.
>
> > Documents integration performance
>
> We conducted additional experiments on the multi-document fusion test set, RGB Integration, which contains answers across multiple documents and effectively tests the model's information fusion capability. The experimental results indicate that without fine-tuning, whether using the composite or reordered approach, directly filling the document KV Cache yields poor performance. After fine-tuning, even with strong correlations between documents, the performance of TurboRAG is comparable to that of Naïve RAG.
>
> | Model                           | ZH Noise 0.2 | ZH Noise 0.4 | ZH Noise 0.6 | ZH Avg. | EN Noise 0.2 | EN Noise 0.4 | EN Noise 0.6 | EN Avg. |
> |---------------------------------|--------------|--------------|--------------|---------|--------------|--------------|--------------|---------|
> | Naïve RAG                       | 50           | 46           | 29           | 42      | 57           | 48           | 36           | 47      |
> | TurboRAG-composite w/o fine-tuning | 35           | 27           | 18           | 27      | 40           | 27           | 27           | 31      |
> | TurboRAG-reordered w/o fine-tuning | 30           | 21           | 20           | 24      | 31           | 23           | 19           | 24      |
> | TurboRAG-composite              | 53           | 41           | 32           | 42      | 58           | 48           | 34           | 47      |
> | TurboRAG-reordered              | 56           | 44           | 32           | 44      | 57           | 51           | 34           | 47      |
>
> > Limitation section
>
> Thank you for your reminder; we have added a discussion on limitations in the new version.
>
> > GPT-4o Chinese performance and TTFT in RGB
>
> Our test set includes both English and Chinese data, and GPT-4o has strong Chinese capabilities, outperforming many open-source models and ranking first on some authoritative Chinese benchmarks, such as https://github.com/jeinlee1991/chinese-llm-benchmark. Additionally, our intention was not to prove that our model is superior to GPT-4o, but rather to provide a benchmark reference for the model's RAG capabilities.
>
> Regarding the RGB English dataset, we have added experimental data, which still shows 2.4x speedup over the Naive RAG. We will include the detailed experiments in the appendix.
>
> | Model      | Context Length (tokens) | TTFT (ms) | Speedup |
> |------------|--------------------------|-----------|---------|
> | Naïve RAG  | 743                       | 87        | 2.42x       |
> | TurboRAG   | 743                      | 36        |    |
>
>
> > Why is gpt-4o's baseline not reported for the LongBench multi-doc QA datasets?
>
> The experiments with TurboRAG primarily demonstrate that using the TurboRAG approach on the same model offers latency advantage over the standard RAG method, without compromising the question-and-answer performance. Given limited resources, we only used a 7B model for the experiments, and our intention was not to compare the model's capabilities with GPT-4o.
>
> > HotpotQA data leakage
>
> Thanks for the comment. Please refer to overall comment C4.
>
> > Experiments for Llama3.1 8B and 70B.
>
> We appreciate the suggestion. Please refer to the overall Comment C3.
>
> > Link to code
>
> We have created an anonymous GitHub repository and uploaded our code. The results presented in the paper can be reproduced by following the guideline provided in the repo. (https://anonymous.4open.science/r/TurboRAG-D732)
>
> > limitation section
>
> Based on your feedback, we have added a limitations section.

---

### Author Response · Authors · 2024-11-28

We thank all reviewers for the time and effort spent on reviewing the paper and providing insightful suggestions on how to consolidate it.

We address a few key common points below for brevity and resolve each detailed comment later. We also carried out a number of additional experiments to help better understand the effectiveness of our method, and anonymously open-sourced the code for community to reproduce the results (https://anonymous.4open.science/r/TurboRAG-D732).

**Common point 1 (C1)**: Reviewers commonly concern about the **storage cost** required by TurboRAG. We used Qwen2 in our paper for evaluation whose KV cache size is about 28M for each document assuming it contains 512 tokens. Even 100K documents only consume 2.8T storage. We admit that the storage cost increases as the model size expands. But we would argue that this is an insignificant cost due to 1) storage is generally very cheap compared to the computation resources (i.e. GPU); 2) exisiting popular LLM models commonly use group-query-attention(GQA) which significantly reduces the kv cache size, thus storage size; 3) prior work, such as RAGCache, also attempted to store kv cache and it might need even more storage space for high cache hit rate.

**Common point 2 (C2)**: Reviewers wanted the **speedup on short documents**. We added new experiments on the RGB benchmark to show the performance of TurboRAG, it turned out that our method can still achieve 2.42x speedup on even such short doc (e.g. containing 743 tokens). Reviewer #3 suggested a comparison against Naive RAG using short docs to measure the H2D overhead. Our initial evaluation on combined sequence length of 256-1024 (i.e. tens to a few hundred tokens per doc) demonstrated that H2D overheads dominate only when the combined sequence length <= 256 and batch size <= 4. It otherwise yields notable performance gain. However, such short sequence length and batch size is impratical and rarely used in any production settings. Please refer to Table 7 and Table 9 in our new version for more detailed results.

**Common point 3 (C3)**: Some comments asked to clarify the **compatibility of TurboRAG to other model architectures**. This is an insightful question. However, due to the resuource limitation, we only performed extensive experiments on Qwen2-7B, which has been widely recognized as one of the powerful open-source LLMs. It has achieved impressive ranking in several well-known benchmarks, such as Open LLM Leaderboard. It shares the similar architecture to other major popular LLM models, e.g. LLaMA, models in which both model commonly adopted ROPE and cross attention algorithms. We do agree that the experiments on other LLM architectures would make the paper more compelling, and we will add some comparsion results with LLaMA3.1-8B in the future revision.

**Common Point 4 (C4)**: **HotpotQA Data Leakage**. In the LongBench paper, it is mentioned that "Since LLMs may have already been trained on the training set of some of our collected public datasets, to avoid test leakage, we extract data from the test sets of these public datasets, with the exception of VCSUM due to its insufficient data in its test set." LongBench actually used the HotpotQA test set. We fine-trained TurboRAG using the HotpotQA training set, it hence wouldn't cause data leakage. Similar approaches can be also found in other previous work, e.g. https://openreview.net/pdf?id=nkOMLBIiI7 and https://arxiv.org/pdf/2404.15420, etc.

---

### Note · Authors · 2025-02-11

I have read and agree with the venue's withdrawal policy on behalf of myself and my co-authors.

---

### Meta-Review · Area_Chair_aj5M · 2024-12-20

**Metareview:**

Retrieval-Augmented Generation (RAG) suffers from significant increase in latency (time-to-first-token (TTFT)). This paper proposes to store KV cache of a retrieval corpus offline and retrieve them at inference time, significantly improving the latency. Results on the RGB benchmark and LongBench multi-document QA show up to 9.4x speedup.

Strengths
- The paper introduced a new technique that leads to significant speedup (8keK, 7nok, KBLU)
- It is well-motivated and the paper is well written (8keK, 7nok, KBLU)
- Empirical results are very strong (8keK, 7nok, KBLU)

Weaknesses
- Storage cost as a cost in saving latency is too significant. The authors mentioned that 100K documents consume 2.8T storage, and 1M documents consume 28T storage. Given that most retrieval corpus like Wikipedia is larger than 1M, the storage requirement is likely to be impractical (8keK, 7nok, VvUh).
- Insufficient discussion / Lack of empirical comparison to work that sparsify attention and/or improve efficiency of RAG/long-context LM (8keK, 7nok)
- Lack of discussion on scalability, such as the impact of retrieval corpus as it would impact both storage requirement and latency (as similarity search will also be more expensive) (7nok)

**Additional Comments On Reviewer Discussion:**

Weaknesses addressed during rebuttal
- Difference from RAGCache (VvUh)
- Clarification on a few details on metrics and ablations (KBLU)

---

### Decision · Program_Chairs · 2025-01-22

Reject